# Microneedles: One-Plane Bevel-Tipped Fabrication by 3D-Printing Processes

**DOI:** 10.3390/molecules27196634

**Published:** 2022-10-06

**Authors:** Isabella Villota, Paulo C. Calvo, Oscar I. Campo, Faruk Fonthal

**Affiliations:** 1Biomedical Engineering Research Group—GBIO, Universidad Autónoma de Occidente, Cali 760030, Colombia; 2Science and Engineering of Materials Research Group—GCIM, Universidad Autónoma de Occidente, Cali 760030, Colombia

**Keywords:** microneedles, transdermal drug delivery, finite element analysis, 3D printing

## Abstract

This article presents microneedles analyses where the design parameters studied included length and inner and outer diameter ranges. A mathematical model was also used to generalize outer and inner diameter ratios in the obtained ranges. Following this, the range of inner and outer diameters was completed by mechanical simulations, ranging from 30 μm to 134 μm as the inner diameter range and 208 μm to 250 μm as the outer diameter range. With these ranges, a mathematical model was made using fourth-order polynomial regressions with a correlation of 0.9993, ensuring a safety factor of four in which von Misses forces of the microneedle are around 17.931 MPa; the ANSYS software was used to analyze the mechanical behavior of the microneedles. In addition, the microneedle concept was made by 3D printing using a bio-compatible resin of class 1. The features presented by the microneedle designed in this study make it a promising option for implementation in a transdermal drug-delivery device.

## 1. Introduction

Transdermal drug delivery (TDD) is a minimally invasive drug that passes through different skin layers, resulting in painless and comfortable drug administration [1,2]. This method is becoming increasingly appealing and desirable for drug delivery because of its advantages compared to conventional drug-delivery methods [1,2,3,4,5].

Despite being a great drug delivery alternative, TDD faces some challenges, of which one of the most important is that not all drugs can pass through the skin with the necessary characteristics for therapeutic actions [6]. Currently, methods are used to improve the transdermal delivery of insulin and other drugs using new micro- and nanotechnologies [7,8,9] that help overcome the challenges of TDD.

On the other hand, there are different devices for TDD, such as the transdermal patch composed of microneedles (MNs). Several researchers have studied MNs as a tool for transdermal drug deliveries. In 2019, a research group presented a review focusing on the MN approach’s potential for transdermal drug-delivery systems. Waghule et al., in 2019, established a range of dimensions for MN; the sizes are in the order of microns, mostly with lengths of 150 μm up to 1500 μm, this being sufficient for releasing the drug into the epidermis, with widths between 50 μm and 250 μm [6]. Economidou et al., in 2021, fabricated hollow cone-shaped MNs with a base diameter of 1000 µm and a tip diameter of 100 µm [7]. Rogkas et al., in 2022, presented improvements made with respect to MNs in their microscale dimensions using the additive manufacturing technique [10]. Drug delivery using MN devices has shown good permeability and efficacy [11].

There are different techniques for manufacturing MNs, especially microelectromechanical systems technologies [12], most of which require tools with high costs. As a result, various researchers used alternative methods that are less costly and sometimes easier to implement. One of these methods is additive manufacturing (3D printing), which has revolutionized pharmaceutical and biomedical sciences due to the manufacturing speed and cost-effective prototypes [13]. Three-dimensional printing makes using different materials possible. In recent years, several researchers obtained promising results in studies that have been used for MN printing [14,15,16]; these studies show that there is still a long way to go with respect to this method. In addition, other authors who have used this method to produce MNs used biocompatible resins [16,17,18].

This paper presents the proposed designs of two different shapes of MNs based on the tip structure of one- and three-bevel hypodermic needles, with a relatively simple form that can facilitate their implementation in 3D printing with class 1 biocompatible resin that has high mechanical strength and puncture capacity. Finite element analysis (FEA) makes it possible to observe and validate the mechanical behavior of loads applied to each MN structure. A mathematical model is also obtained that relates the inner and outer diameter of the MN within established ranges of these design parameters. Three-dimensional printing of the two designs was also implemented in the Form2 (Formlabs) printer to observe which methods achieve the best print resolutions.

## 2. Theoretical Analysis

### 2.1. Structural Design

The structural design of MNs is based on creating a one-plane bevel-tipped needle and three bevel tips. These structures are relatively simple and allow an adequate load distribution on the network, avoiding structural damage during the MN insertion process.

The basic shape of the MNs is a cylinder with a length of 450 μm. The inner and outer diameter ranges are established and any value can be used for them because the dimensions for manufacturing MNs greatly depend on available resources. The ranges maintain the values of inner and outer diameters without exceeding the sizes of an MN without affecting the mechanical behavior of the MN, and they constitute a good safety factor for future applications in the medical area.

In the inner diameter range, a minimum of 30 μm is established and a maximum of 250 μm is established for the outer diameter. If this value is outstripped, it would be outside the accepted dimensions for an MN [6]. Mechanical simulations establish that this diameter range ensures that the MN can overcome the force necessary to pierce the skin without presenting mechanical failures, providing a safety factor of 4. Values similar and close to this inner diameter range have been used in medical device designs, such as needles, that prove suitable for utilization [19,20].

For the manufacture of MNs, the printing quality will determine the diameter values used to obtain a good structure with good printing resolutions.

#### 2.1.1. Microneedle with the Tip in the Form of a One-Plane Bevel-Tipped Needle (MNTB1)

The structure of this MN has a tip forming an angle (*θ*) of 45° between the lateral surface and the bevelled plane, a length of 450 μm and a range of inner diameter (*β*) and outer diameter (*α*). A single-plane bevel hypodermic needle inspired the shape of this tip, and the end acts as a cutting edge (see Figure 1) bevelled.

This MN generates a force asymmetrically, causing the tissue cut to occur at a compensating angle depending on the bevel angle, needle’s flexibility, and tissue properties [21].

#### 2.1.2. Microneedle with the Tip in the Form of a 3-Bevel Tip (MNTB3)

The MN structure of the tip is inspired by a hypodermic needle three bevel tip; i.e., it has three areas where the force exerted by the skin will be distributed during penetration, and it has two shear angles, *θ_1_* = 30° and *θ_2_* = 35°, as shown in Figure 2, a length of 450 μm and a range of inner diameter (*β*) and outer diameter (*α*).

The shape of the tip that is formed by the uppercuts (bevel 2 and 3) induces a sharper tip for this MN, which is a fundamental characteristic of the performance of needle insertions because the contact area between the MN is mainly concentrated at the part of the pointed tip at the time of the initial cut [22].

### 2.2. Theoretical Microneedle Mechanical Analysis

The stratum corneum is the principal cutaneous layer that mainly influences the viscoelastic property of the skin; this property is responsible for generating resistance when the skin is perforated, providing the maximum insertion resistance of an MN [23] and causing compressions on the MN structure. It is necessary to overcome the skin’s opposition for the MN to penetrate the skin.

The evaluation of MN mechanical stresses simulates the forces to which it is subjected to at the moment of insertion; this is where the MN is exposed to maximum resistance by the skin.

The axial force that the MN can withstand without producing irreversible negative effects on its structure is provided with the compressive stress or yield stress (*σ_y_*); this stress is described by Equation (1) and relates to the cross-section (*A*) of the analyzed section with the force that the MN can withstand without breaking (*F_compressive_*).
(1)$$\sigma_{y}=\frac{F_{compressive}}{A},\;$$

It is necessary to know the mechanical properties of the material used in order to study the forces on the MN, which is a class 1 biocompatible resin. These material properties include a tensile strength of 73 Mpa and a Young’s modulus of 2.9 Gpa since the values obtained in the results depend directly on these properties.

In addition, the *F_compressive_* used was 0.5 N. This value is adequate for simulating the force needed to pierce the skin in hydration conditions because it is within ranges established in other studies [24,25].

## 3. Finite Element Analysis (FEA)

The main reason for the mechanical analysis of MNs is to determine the behavior of the structure and the material chosen because they must be able to correctly support the skin’s resistance without generating adverse effects, such as buckling in the MN structure during the insertion process. A fundamental factor that decreases the probability of buckling is an adequate wall thickness.

The simulations of the two-needle designs performed with the biocompatible resin class 1, with the mechanical properties mentioned above and a Poisson coefficient of 0.35 established by resins with similar characteristics, were used [26,27,28].

The results obtained with these simulations show that the MN’s design presents lower von Misses stresses and deformations [24]. Other results are the values of the available inner and outer diameter ranges used in mechanical simulations. With these, a mathematical model is produced and it generalizes the relationship of these two design parameters within the ranges obtained, ensuring the mechanical integrity of the MN within these ranges and also allowing the MN to have an adequate safety factor for its future medical implementation.

### 3.1. Initial Mechanical Analysis of Microneedle Structures

For this mechanical analysis, the two MN designs (MNTB1 and MNTB3) were evaluated under the same conditions, a compression force of 0,5N and specific geometric dimensions: MN length of 450 μm, inner diameter (*β*) of 30 μm and outer diameter (*α*) of 250 μm, values within the initially known ranges.

These simulations showed that the MN’s design with the lowest von Misses stresses was MNTB1, with a maximum stress of 12.451 Mpa (see Figure 3) and a maximum strain of 0.0015442, while the MNTB3 design presented a maximum stress of 15.903 MPa (see Figure 4) and a maximum deformation of 0.0013586.

### 3.2. Generalized Outer and Inner Diameter Ranges for MNTB1 Designs with a Factor of Safety of 4

The generalized ranges were obtained for the MN design with the lowest von Misses stresses, MNTB1.

With a minimum diameter of 30 μm, the outer diameter varied in the mechanical simulation until finding the value where the von Misses forces were as close as possible to 18.250 MPa (since this is the permissible value for safety factor 4). Considering that the material’s tensile strength is 73 MPa, the minimum outer diameter was around 208 μm, the maximum von Misses stress was 17.818 MPa (see Figure 5) and the maximum outer diameter was 250 μm; the same procedure was performed. However, the inner diameter varied until finding the maximum value (where the MN did not exceed the tensile strength of the resin, which is 134 μm), the maximum inner diameter and a von Misses stress of 17.931 MPa (see Figure 6). Table 1 shows the values obtained for the inner and outer diameter.

### 3.3. A Mathematical Model for Generalizing the Outer/Inner Ratio Diameter within the Obtained Ranges

A mathematical model that generalizes the relationship of the diameters within the ranges obtained was developed. One characteristic of the model is flexibility in the microneedle’s design since any diameter value within the ranges can be used; this can be very useful when manufacturing the microneedle. It also ensures that the MN’s design has an adequate safety factor for future manufacturing and implementation. On many occasions, the available manufacturing resources may limit microneedles’ manufacture in micrometer sizes, as specialized technology is required.

First, a graph was made to observe the behavior and trend of the data, for which 12 values were chosen within the inner and outer diameter ranges (see Table 2). This table also shows that none of the values in this range presented stresses higher than the permissible value for a safety factor of 4, confirming that the design parameters preserved the mechanical integrity of the MN in these ranges. Next, we used fourth-degree polynomial regressions, which included the one that perfectly fitted the trend of the data (see Figure 7). Finally, a mathematical model corresponding to the form shown in equation two was obtained for the study.
(2)$$Dext=Ad^{4}+Bd^{3}+Cd^{2}+Dd+E,\;$$

Here, *A* presents a value of 4.9622 × 10^−7^, *B* of −1.8167× 10^−4^, *C* of 0.0256, *D* of −1.1671 and *E* of 224.3235, which are values found in MATLAB when performing polynomial regressions. Moreover, a data correlation of 0.9993 was obtained, which indicates that the behavior of the data and the fourth-degree polynomial regression have a very high and good correlation.

The inner or outer diameter values found by this mathematical model will allow the design to present a von Misses stress similar to or close to 17.998 MPa, which is the average value of the data obtained in this study.

## 4. Microneedle 3D Printing

### 4.1. 3D-Printing Process

The 3D printing of MNs was performed with the Form 2 (Formlabs, GoPrint3D, Ripon, North Yorkshire, UK)—and a class 1 biocompatible resin (see Figure 8A). For this process, a printing angle of 0° concerning the printing platform was configured. Once the impression was finished, it was necessary to immerse the piece with the MN in 99% isopropyl alcohol (see Figure 8B), and then it was placed in an ultrasonic cleaner for 10 min to eliminate the excess resin generated after the printing process (see Figure 8C). At the end of this process, the MN was left to dry for 20 min and then cured in the FormCure (Formlabs, GoPrint3D, Ripon, North Yorkshire, UK) for 30 min at 60 °C according to the material protocol (see Figure 8D).

### 4.2. 3D Manufacture of MNTB1

The 3D printing of MNs was performed with the Form 2 (Formlabs, GoPrint3D, Ripon, North Yorkshire, UK) and a type I biocompatible resin. FormLabs commercial biocompatible resin was used for manufacturing. The biocompatible tests showed that it is not cytotoxic and stable against physical–chemical processes [29]. This resin type has been widely used in surgical guides, drilling templates, pilot drill guides and device sizing templates [29].

The resin is considered a Type I medical device according to the Medical Device Directive (93/42/EEC) in the EU and Section 201 (h) of the Federal Food Drug and Cosmetic (FD&C) Act; the mechanical, light-curing processes are detailed in [29] by the manufacturer. The cleaning and curing processes of the MNs were performed following the protocols given by the material manufacturer.

The MNTB1 design was manufactured in 3D printing, and a good printing resolution was observed. The printed format did correspond to the modeled design and that the experimental geometric dimensions were very close to the theoretical ones. This type of resin has been used in several studies in which the biocompatibility and stability of the material are important, and it has been concluded that it can become useful as a drug delivery route [30]; however, the manufacturer does not know the material’s properties in the presence of specific drugs, which suggests that future studies should evaluate this characteristic.

Two printed-out MNs were fabricated: one with the minimum values and the other with the maximum diameters. Although the impression achieved a physical appearance that is very similar to the proposed design, resin clogged in the inner holes of the MN at both minimum and maximum diameter values. Still, for the maximum values, less clogging occurred. As shown in Figure 9, an improved printing resolution was obtained and larger dimensions resulted.

## 5. Conclusions and Future Outlook

This research paper shows useful and relevant information for designing and fabricating MN, analyzing the mechanical behavior of the structures designed and finding the ranges of inner and outer diameters necessary so that the MN does not induce mechanical failures and that these dimensions were permitted for 3D printing. A mathematical model was also found to generalize the outer and inner diameter ratios in the obtained ranges.

The FEA simulations showed that the MNTB1 design was the one that obtained the lowest von Misses stresses, and this design was then the one with the best mechanical behavior. In addition, simulations showed that the design, dimensions, and material support the force exerted by the skin at the time of penetration of the MN, which shows that it can be coupled in a method of transdermal drug delivery. Moreover, the inner and outer diameter ranges allow the needle to behave mechanically and appropriately during insertion. The range is between 30 μm and 134 μm for the inner diameter and 208 μm to 250 μm for the outer diameter.

The mathematical model that generalizes the relation between the inner and outer diameters permits a universal design in manufacturing because there is no single value for the diameters. The model does not fail mechanically and can be printed.

The fabrication of the MNTB1 model in 3D printing showed high resolutions of impressive appearance and dimensions very similar to those of the designed model. However, the inner cavity of the MN presented resin clogging, as the MNs are in the order of micrometers, inducing a more rigorous 3D-printing process. Three-dimensional printing still faces many challenges and must be studied more extensively to perfect this method, especially with respect to small details such as inner holes.

In future studies, we hope to continue to contribute to the field of transdermal drug delivery; to realize this, it is necessary to complement the results found in this study. Therefore, we will perform tests with manufacturing processes and materials that are different from those explored in this paper in order to make comparisons between the results obtained from this proposed design and to observe if another material or manufacturing process also has compromising scopes similar to the one explored at this stage. On the other hand, at a more advanced stage, research studies carrying out clinical research to support future implementations of the designed microneedles in a transdermal drug delivery system are expected.

## Figures and Tables

**Figure 1 molecules-27-06634-f001:**
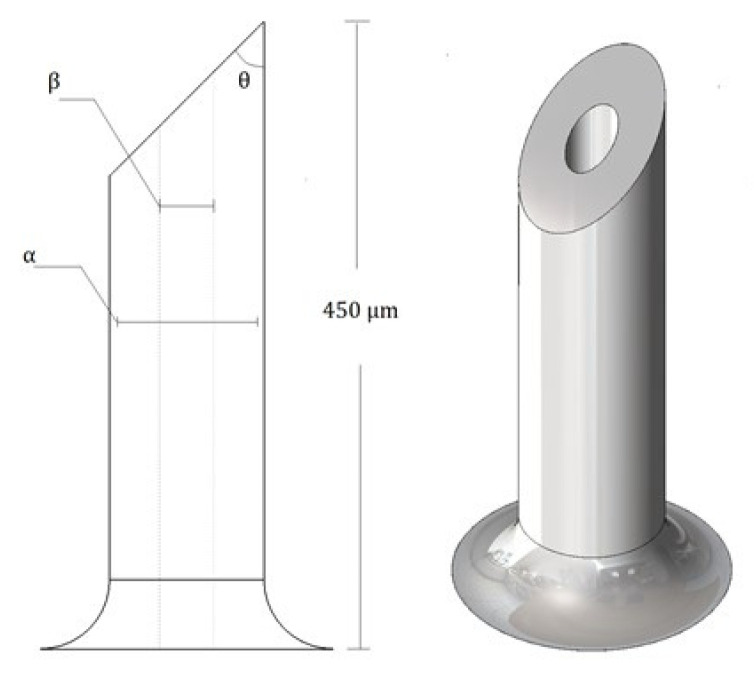
Schematic illustration of MNTB1 design.

**Figure 2 molecules-27-06634-f002:**
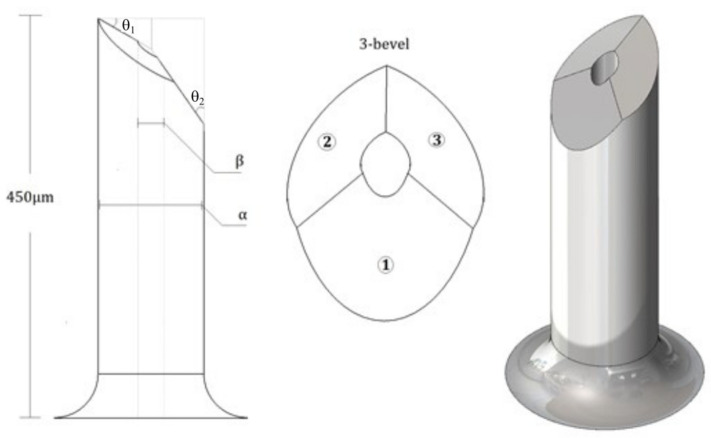
Schematic illustration of MNTB3 design.

**Figure 3 molecules-27-06634-f003:**
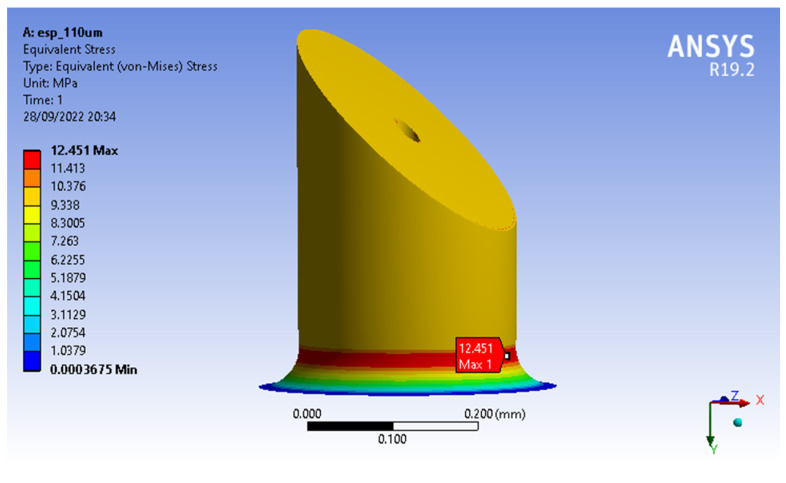
Von Misses Stresses MNTB1.

**Figure 4 molecules-27-06634-f004:**
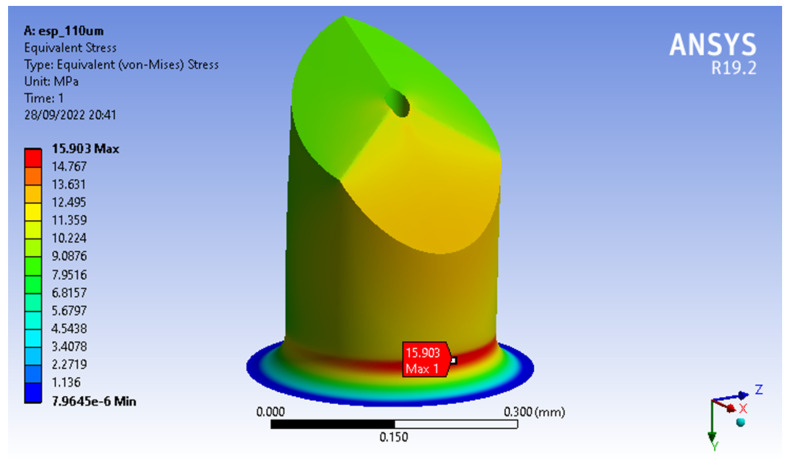
Von Misses Stresses MNTB3.

**Figure 5 molecules-27-06634-f005:**
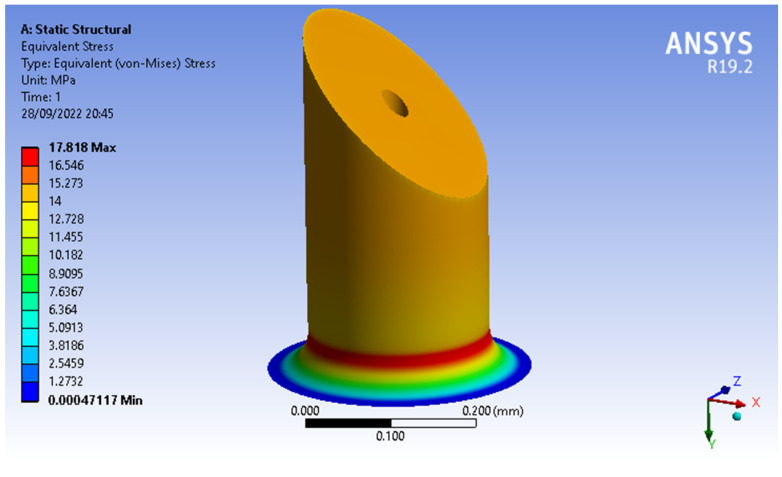
Von Misses stresses of MNTB1: an inner diameter of 30 μm.

**Figure 6 molecules-27-06634-f006:**
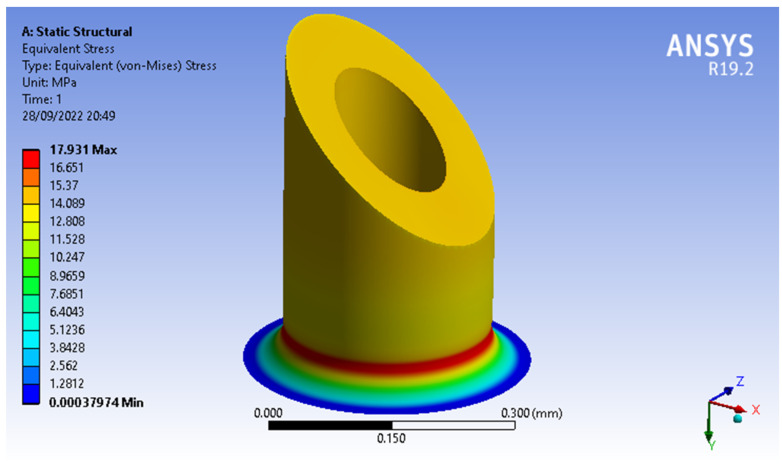
Von Misses stresses of MNTB1: an inner diameter of 134 μm.

**Figure 7 molecules-27-06634-f007:**
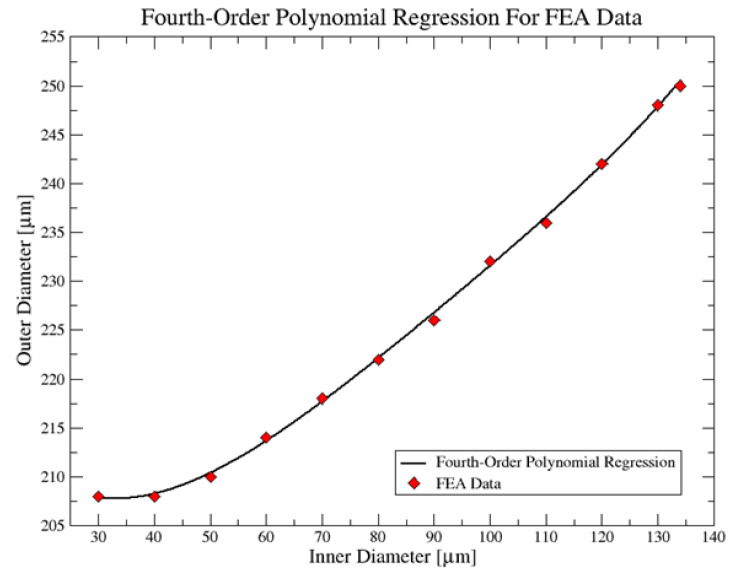
Graph of the fourth-degree polynomial relationship: The x-axis is the inner diameter (µm), and the y-axis is the outer diameter (μm).

**Figure 8 molecules-27-06634-f008:**
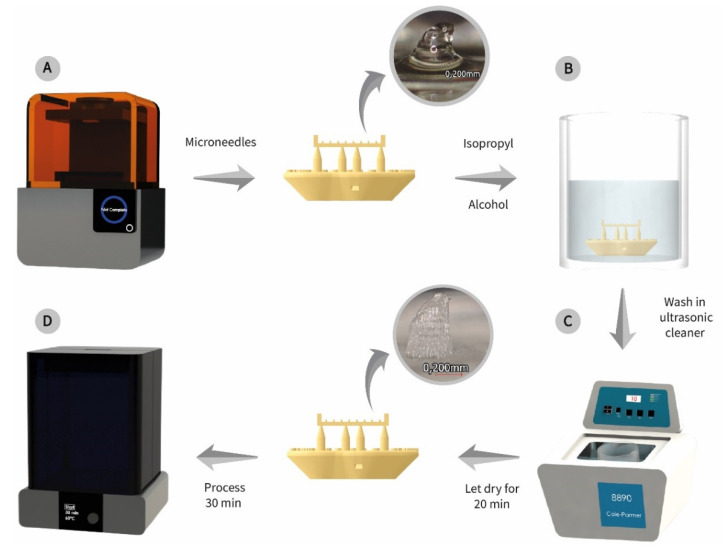
Microneedle 3D-printing process. (**A**) Microneedle fabrication in the Form 2 printer, (**B**) the immersed piece in isopropyl alcohol, (**C**) ultrasonic cleaner to eliminate the excess resin generated and (**D**) cured piece in the FormCure at 60 °C.

**Figure 9 molecules-27-06634-f009:**
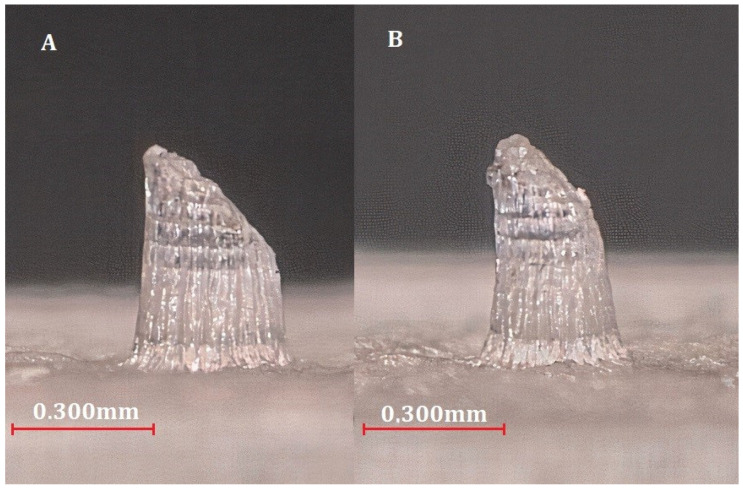
Printing of the MNTB1 design: (**A**) printing with maximum diameter values; (**B**) printing with minimum diameter values.

**Table 1 molecules-27-06634-t001:** Result of inner and outer diameter ranges.

Diameter (μm)	Min. Value	Max. Value
Inner	30	134
Outer	208	250

**Table 2 molecules-27-06634-t002:** Inner and outer diameter values within the established range and von Misses stress values were obtained in each case.

Inner Diameter (μm)	Outer Diameter (μm)	Von Misses Stresses (MPa)
30	208	17.818
40	208	18.043
50	210	18.170
60	214	18.023
70	218	17.976
80	222	17.983
90	226	18.080
100	232	17.860
110	236	18.090
120	242	18.023
130	248	17.979
134	250	17.931

## Data Availability

Not applicable.

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
