# Peer review of "Microneedles: One-Plane Bevel-Tipped Fabrication by 3D-Printing Processes"

_molecules, 2022, doi:10.3390/molecules27196634_

Round 1

Reviewer 1 Report

This manuscript reports a study about 3D-printed microneedles. Specifically, the author discussed the design parameters including length, inner and outer diameter range. Overall, this report presents a comprehensive understanding about microneedles. However, some details deteriorated the technical merit of this work. Specific comments are provided below for authors’ consideration to improve this manuscript before its publication.

1.       An abstract is supposed to highlight the importance of your work, briefly summarize your design, the most important results, and potential influence, instead of listing the research method. The author may need to revise the abstract accordingly.

2.       An introduction is supposed to address the problem and the novelty of your design. For example, it could be more precise by comparing the advantages of your design with others’ work.

3.       The author should provide an individual section to disclose the methods and details of theoretical analysis and 3D printing. Some examples are provided below:

Macromolecules 2017, 50, 17, 6637–6646

Polymer 237 (2021) 124323

4.       The author should provide control groups to demonstrate the improvement of the new design.

5.       To provide more detailed characterization and analysis in 3D printing besides two pictures to echo the theoretical analysis. The two photos in Fig. 8 were from two different angle, thus they are not comparable. Also, to demonstrate better resolution, the pictures should at least include the needles at a different angle or from top/side/bottom 

Author Response

Dear Reviewer.

We want to thank you for your time in helping us improve our manuscript. You are correct; for that reason, we have made the following corrections to the document:

  • An abstract is supposed to highlight the importance of your work and briefly summarize your design, the most important results, and potential influence instead of listing the research method. The author may need to revise the abstract accordingly.

Response #1, The abstract was modified as suggested by the reviewer and corrected in the document. "This article presents the microneedles analysis where the design parameters studied were length and inner and outer diameter range.  
A mathematical model was also found to generalize the outer and inner diameter ratios in the obtained ranges. Following this, the range of inner and outer diameters was completed by mechanical simulations, getting from 30 μm to 134 μm as the inner diameter range and 208 μm to 250 μm as the outer diameter range. With these ranges, a mathematical model was made using fourth-order polynomial regression, with a correlation of 0,9993, ensuring a safety factor of four, where Von Misses forces of the Microneedle are around 17.931 MPa; the ANSYS software was used to analyze the mechanical behavior of the microneedles. In addition, the microneedle concept was made by 3D printing using a bio-compatible resin of class 1. The features presented by the Microneedle designed in this work make it a promising option to be implemented in a transdermal drug delivery device."

  • An introduction is supposed to address the problem and the novelty of your design. For example, it could be more precise by comparing the advantages of your design with others' work.

Response #2, the introduction was modified as suggested by the reviewer and corrected in the document. We have changed the third paragraph of the introduction to highlight the importance of the dimensions, improving the explanation of the potential of the results presented in the document. "On the other hand, there are different devices for TDD, such as the transdermal patch composed of microneedles (MN). Several researchers have studied MNs as a tool for transdermal drug delivery. In 2019, a research group presented a review focusing on the MN approach's potential for transdermal drug delivery systems. Waghule et al., in 2019, established the ranges of dimensions for MN; the sizes are in the order of microns, mostly with lengths of 150 μm up to 1500 μm, this being sufficient to release the drug into the epidermis, with width between 50 μm and 250 μm [6]. Economidou et al., in 2021, fabricated hollow cone-shaped MNs with a base diameter of 1000 µm and a tip diameter of 100 µm [7]. Rogkas et al., in 2022, presented the improvements made to MN in their microscale dimensions using the additive manufacturing technique [10]."

  • The author should provide an individual section to disclose the methods and details of theoretical analysis and 3D printing. Some examples are provided below:

Response #3, the document was improved and included a section that explained the method used and the theoretical analysis. We were included a Fig 8.

  1. Microneedle 3D Printing

4.1. 3d printing process

The 3D printing of the MNs was performed with the Form 2- Formlabs printer and a class 1 biocompatible resin, see fig 8A. For this process, a printing angle of 0º concerning the printing platform was configured. Once the impression was finished, it was necessary to immerse the piece with the Microneedle in 99% isopropyl alcohol, see fig 8b, and place it in an ultrasonic cleaner for 10 minutes to eliminate the excess resin generated after the printing, see fig 8c. At the end of this process, the MN was left to dry for 20 minutes and then cured in the FormCure for 30 minutes at 60°C, according to the material protocol, see fig 8D.

  • The author should provide control groups to demonstrate the improvement of the new design.

Response #4, Currently, a clinical study has not been carried out where we include the control requested by the evaluator, but we have added it as future work. “In future work, we hope to continue to contribute to the field of transdermal drug delivery, and to do so, it is necessary to complement the study for this work. Therefore, we will perform tests with manufacturing processes and materials different from those explored in this paper to compare the results obtained from this proposed design and observe if another material or manufacturing process also has compromising scopes like the one explored at this stage. On the other hand, at a more advanced stage, it is expected to be able to carry out clinical research to support future implementation of the designed microneedles in a transdermal drug delivery system.”

  • To provide more detailed characterization and analysis in 3D printing besides two pictures to echo the theoretical analysis. The two photos in Fig. 8 were from two different angle, thus they are not comparable. Also, to demonstrate better resolution, the pictures should at least include the needles at a different angle or from top/side/bottom.

Response #5, The reviewer is right about the image quality of Fig. 8, which has been modified and included in the document as Fig. 9.

Reviewer 2 Report

The manuscript presents the obtaining of microneedles by the 3D printing process. The mechanical properties of microneedles were determined. By means of a mathematical model, the inner diameter was related to the outer diameter of the microneedles. It was determined that microneedles can be used for controlled drug release. The evaluator finds little novelty in the work and the evaluator believes that the manuscript can be improved.

Author Response

Dear reviewer.

Thank you very much for your comments.

Reviewer 3 Report

The current article is interesting and provide novel data about the application of drug delivery with very suitable presentation, but some points are needed:

1- If authors put graphical diagram for summarizing all of the procedures of 3D print , it will be scientifically clear for readers.

2-Live images or clinical investigation is recommended or even put more information with statistical analysis.

3-High resolution images are recommended for Fig.8.

Author Response

Dear Reviewer.

We want to thank you for your time in helping us improve our manuscript. You are correct; for that reason, we have made the following corrections to the document:

  • If authors put graphical diagram for summarizing all of the procedures of 3D print , it will be scientifically clear for readers.

Response #1, the document was improved and included a section that explained the method used and the theoretical analysis, and included a figure summary to explain better the MN manufacture, Fig 8.

  1. Microneedle 3D Printing

4.1. 3d printing process

The 3D printing of the MNs was performed with the Form 2- Formlabs printer and a class 1 biocompatible resin, see fig 8A. For this process, a printing angle of 0º concerning the printing platform was configured. Once the impression was finished, it was necessary to immerse the piece with the Microneedle in 99% isopropyl alcohol, see fig 8b, and place it in an ultrasonic cleaner for 10 minutes to eliminate the excess resin generated after the printing, see fig 8c. At the end of this process, the MN was left to dry for 20 minutes and then cured in the FormCure for 30 minutes at 60°C, according to the material protocol, see fig 8D.

  • Live images or clinical investigation is recommended or even put more information with statistical analysis.

Response #2, currently, a clinical study has not been carried out where we include the control requested by the evaluator, but we have added it as future work. “In future work, we hope to continue to contribute to the field of transdermal drug delivery, and to do so, it is necessary to complement the study for this work. Therefore, we will perform tests with manufacturing processes and materials different from those explored in this paper to compare the results obtained from this proposed design and observe if another material or manufacturing process also has compromising scopes like the one explored at this stage. On the other hand, at a more advanced stage, it is expected to be able to carry out clinical research to support future implementation of the designed microneedles in a transdermal drug delivery system.”

  • High resolution images are recommended for Fig.8.

Response #3, the reviewer is right about the image quality of Fig. 8, which has been modified and included in the document as Fig. 9.

Round 2

Reviewer 1 Report

The manuscript has been revised accordingly.

Author Response

Dear reviewer.

Thank you very much for your suggestions and comments to improve the document; thank you for taking the time to evaluate the paper.